# FOXO1 Couples KGF and PI-3K/AKT Signaling to NKX2.1-Regulated Differentiation of Alveolar Epithelial Cells

**DOI:** 10.3390/cells11071122

**Published:** 2022-03-26

**Authors:** Qian Zhong, Yixin Liu, Michele Ramos Correa, Crystal Nicole Marconett, Parviz Minoo, Changgong Li, David K. Ann, Beiyun Zhou, Zea Borok

**Affiliations:** 1Division of Pulmonary, Critical Care and Sleep Medicine, Department of Medicine, Keck School of Medicine, University of Southern California, Los Angeles, CA 90089, USA; zhongqian@sysucc.org.cn (Q.Z.); yixinliu@usc.edu (Y.L.); 2Hastings Center for Pulmonary Research, Keck School of Medicine, University of Southern California, Los Angeles, CA 90089, USA; mramosco@usc.edu (M.R.C.); cmarcone@usc.edu (C.N.M.); minoo@usc.edu (P.M.); changgon@usc.edu (C.L.); 3USC Norris Comprehensive Cancer Center, Keck School of Medicine, University of Southern California, Los Angeles, CA 90089, USA; 4Department of Surgery, Keck School of Medicine, University of Southern California, Los Angeles, CA 90089, USA; 5Department of Biochemistry and Molecular Medicine, Keck School of Medicine, University of Southern California, Los Angeles, CA 90089, USA; 6Department of Pediatrics, Keck School of Medicine, University of Southern California, Los Angeles, CA 90089, USA; 7Department of Diabetes Complications and Metabolism, Beckman Research Institute, City of Hope Medical Center, Duarte, CA 91010, USA; dann@coh.org; 8Division of Pulmonary, Critical Care and Sleep Medicine, Department of Medicine, University of California, San Diego, CA 92037, USA

**Keywords:** FOXO1-NKX2.1 interaction, keratinocyte growth factor (KGF), PI-3K/AKT, alveolar epithelial cell, differentiation, transcription

## Abstract

NKX2.1 is a master regulator of lung morphogenesis and cell specification; however, interactions of NKX2.1 with various transcription factors to regulate cell-specific gene expression and cell fate in the distal lung remain incompletely understood. FOXO1 is a key regulator of stem/progenitor cell maintenance/differentiation in several tissues but its role in the regulation of lung alveolar epithelial progenitor homeostasis has not been evaluated. We identified a novel role for FOXO1 in alveolar epithelial cell (AEC) differentiation that results in the removal of NKX2.1 from surfactant gene promoters and the subsequent loss of surfactant expression in alveolar epithelial type I-like (AT1-like) cells. We found that the FOXO1 forkhead domain potentiates a loss of surfactant gene expression through an interaction with the NKX2.1 homeodomain, disrupting NKX2.1 binding to the *SFTPC* promoter. In addition, blocking PI-3K/AKT signaling reduces phosphorylated FOXO-1 (p-FOXO1), allowing accumulated nuclear FOXO1 to interact with NKX2.1 in differentiating AEC. Inhibiting AEC differentiation in vitro with keratinocyte growth factor (KGF) maintained an AT2 cell phenotype through increased PI3K/AKT-mediated FOXO1 phosphorylation, resulting in higher levels of surfactant expression. Together these results indicate that FOXO1 plays a central role in AEC differentiation by directly binding NKX2.1 and suggests an essential role for FOXO1 in mediating AEC homeostasis.

## 1. Introduction 

The lung alveolar epithelium consists of two distinct AEC populations, type I (AT1) and type II (AT2) cells. AT2 cells synthesize and secrete surfactant proteins (SFTPA, SFTPB, SFTPC, and SFTPD), which reduce surface tension at the air-liquid interface of the lung, preventing airspace collapse [1]. AT1 cells, the site of gas exchange, cover > 95% of the alveolar surface, and until recently, have been viewed as terminally differentiated [2]. AT2 cells serve as primary progenitors of the alveolar epithelium, both self-renewing and giving rise to AT1 cells to restore the alveolar epithelium during normal maintenance and repair following lung injury [3,4]. In vitro and in vivo studies have also reported that under certain conditions AT1 cells can give rise to AT2 cells, indicating greater plasticity than previously thought [5,6,7]. Impaired AT2 cell proliferation and differentiation to AT1 cells results in failure of regeneration and aberrant persistence of intermediate cell states in diseases such as idiopathic pulmonary fibrosis (IPF) [8,9,10,11,12]. Several signaling pathways have been reported to promote AT2 cell proliferation. These include fibroblast growth factor (FGF) [6,13,14,15,16,17], Wnt/*β*-catenin [18,19,20], Yes-associated protein/Transcriptional coactivator with PDZ-binding motif (YAP/TAZ) [21,22,23], forkhead box protein M1 (FoxM1) [24], and the ETS family transcription factor 5 (ETV5) [25]. Additionally, Wnt/*β*-catenin [26,27,28,29], Hippo/YAP [7,30,31,32], Notch [33], bone morphogenetic protein (BMP) [34,35], and transforming growth factor-*β* (TGF-β) [5,11,36,37] are also implicated in the regulation of AT2 to AT1 cell differentiation during steady state tissue maintenance and following injury. However, the complex network of transcription factors (TFs) and signaling pathways regulating this phenotypic transition process is not fully understood.

FOXO1 is a member of the FOXO subfamily of forkhead box (FOX) TFs that includes FOXO1, FOXO3, FOXO4, and FOXO6 [38,39]. FOXO1, FOXO3, and FOXO4 are ubiquitous and FOXO6 is predominantly expressed in the brain. The fox family TFs are defined by a conserved 110 amino acid winged helix DNA-binding domain also known as the forkhead (FK) box [39,40]. FOXO1, also known as Forkhead Homologue in Rhabdomyosarcoma (FKHR), was initially identified as a fusion protein with Pax3 in alveolar rhabdomyosarcoma [41]. FOXO1 activity is tightly regulated by changes in protein expression as well as posttranslational modifications such as phosphorylation that influence subcellular localization, molecular half-life, DNA-binding activity, and protein-protein binding ability [39]. Several protein kinases (e.g., phosphoinositide 3-kinase (PI-3K)/protein kinase B (PKB), also known as AKT) phosphorylate FOXO1, leading to FOXO1/14-3-3 interaction and subsequent cytoplasmic localization and further degradation via the ubiquitin–proteasome pathway [42,43,44,45]. Importantly, effects of FOXO family members on gene transcription are frequently found to be mediated via protein–protein interactions rather than direct DNA binding [46].

FOXO1 can function as either a repressor or activator to impact the expression of genes that are involved in a wide variety of cellular processes including cell proliferation, differentiation, survival, and apoptosis [44,47,48,49]. FOXO1 is required for the maintenance of somatic and cancer stem cells, as well as pluripotency in embryonic stem cells [50,51,52,53]. FOXO1 typically functions to inhibit differentiation in multiple cell types, including progenitor cells. For example, FOXO1 prevents differentiation of mesenchymal cells into adipocytes and osteoblasts [54,55,56], enteroendocrine progenitors into mature β cells [57] andhepatic stellate cells into myofibroblasts [58], and also inhibits endothelial cell differentiation [59]. In contrast, FOXO1 has also been shown to promote the transition from clonal expansion to terminal differentiation during adipocyte differentiation [60], suggesting context-dependent effects. In the lung, FOXO1 has been shown to antagonize FoxM1-dependent endothelial cell proliferation in lipopolysaccharide-induced acute lung injury [61]. Although FOXO1 is involved in injury-induced apoptosis in bronchial epithelial cells and AEC [62], a role for FOXO1 in the regulation of alveolar epithelial progenitor homeostasis during normal physiological turnover and injury conditions has not been reported to date.

In contrast to FOXO1, NKX2.1, a homeodomain-containing TF, has been well studied for its role in the homeostasis of lung epithelial cells including AEC. NKX2.1 is essential for lung development and a critical determinant of both proximal and distal lung epithelium-specific gene expression [63,64]. There is strong evidence demonstrating that NKX2.1 activates surfactant gene expression through an interaction with a number of transcriptional co-activators (e.g., GATA-6, NFI, TAZ, and Erm) [65,66,67,68,69]. Recent studies have also found that NKX2.1 is expressed in AT1 cells, binds to and regulates AT1 cell-specific genes, and promotes the diametrically opposed AT1 and AT2 cell fates through differential DNA binding and interaction with transcriptional cofactors (e.g., YAP/TAZ) [31,70]. It is likely that an interacting network of TF, including NKX2.1, regulates phenotypic transitions between AT2 and AT1 cells, but the specific TF interactions with NKX2.1 that leads to differential regulation of gene expression in AT1 vs. AT2 cells have not been elucidated.

In this study, we investigated cell-specific interactions of FOXO1 with NKX2.1, using a two-dimensional in vitro model of rat AEC differentiation. We report that FOXO1 interaction with NKX2.1 results in the loss of transcriptional activation of AT2 cell-specific genes (*Sftpb* and *Sftpc*) suggesting that FOXO1 promotes AT2 to AT1 cell differentiation. Importantly, we found that keratinocyte growth factor (KGF, also known as FGF7), that we and others have previously reported to maintain the AT2 cell phenotype [6,71], regulates SFTPC expression through PI-3K/AKT-dependent phosphorylation of FOXO1. Our results establish a novel mechanism whereby KGF and PI-3K/AKT-mediated cell-specific regulation of FOXO1 phosphorylation modulates the interactions of FOXO1 with NKX2.1 to determine cell-specific gene expression and regulate AEC differentiation.

## 2. Materials and methods

### 2.1. Preparation and Treatment of Primary rat AEC Monolayers

AT2 cells were isolated from ~125 to 150 g adult male Sprague-Dawley rats by enzymatic disaggregation with elastase (2.0–2.5 U/mL; Worthington Biochemical, Freehold, NJ), followed by differential adherence on IgG-coated bacteriological plates as previously described [72]. The cells were suspended in a minimal defined serum-free medium (MDSF) consisting of Dulbecco’s modified Eagle’s medium (DMEM) and Ham’s F12 nutrient mixture in a 1:1 ratio (DMEM-F12), that was supplemented with 1.25 mg/mL bovine serum albumin (BSA), 10 mM HEPES, 0.1 mM non-essential amino acids, 2.0 mM glutamine, 100 U/mL sodium penicillin G and 100 µg/mL streptomycin, and seeded onto tissue culture-treated polycarbonate filters (Transwell, 0.4 µm pore size, Corning Costar, Cambridge, MA, USA) at a density of 10^6^ cells/cm^2^. AT2 cell purity (>90%) was assessed by staining the freshly isolated cells with P180 lamellar membrane protein antibody with a dilution ratio of 1:1000 (Covance Research, Berkeley, CA, USA). Animal protocols were approved by the Institutional Animal Care and Use Committee at the University of Southern California.

### 2.2. Culture of MLE-15 Cells and Nthy-ori 3-1 Cell Line

MLE-15 cells (Dr. J. Whitsett, University of Cincinnati) were cultivated in HITES medium (RPMI 1640 medium (Invitrogen, Carlsbad, CA, USA) that was supplemented with 10 nM hydrocortisone, 5 μg/mL insulin, 5 μg/mL human transferrin, 10 nM *β*-estradiol, 5 μg/mL selenium, 2 mM glutamine, 10 mM HEPES, 100 U/mL penicillin, 100 μg/mL streptomycin, and 4% fetal bovine serum (FBS) (Atlanta Biologicals, Lawrenceville, GA, USA)). The cells were passaged at 70% confluence. The Nthy-ori 3-1 cell line (#90011609, Millipore, Sigma, Burlington, MA, USA) was cultured in RPMI 1640 medium that was supplemented with 2 mM glutamine and 10% FBS.

### 2.3. Plasmids

3.7-*SFTPC*-Luc contains the 3.7-kb human *SFTPC* promoter in pGL2Basic (Promega, Madison, WI, USA). pRC/CMV/NKX2.1 contains the 2.3-kb human *NKX2.1* cDNA in pRC/CMV (Invitrogen). p318 mu-*Sftpc*-Luc contains −318 to -118 of the murine *Sftpc* promoter cloned into SmaI/XhoI sites of pGL2Basic [73]. pCMV6-XL4/FOXO1 containing the human FOXO1 cDNA was purchased from Origene (Rockville, MD, USA). pCDNA3/FOXO1-AAA and pCDNA3/FOXO1 H215R were purchased from Addgene (Cambridge, MA, USA). pCDNA3/FOXO1, pCDNA3-FOXO1-N (1–257 a.a.), pCDNA3-FOXO1-C (211–416 a.a.), pGEX-KG-FOXO1 and pGEX-KG-FOXO1-N (1–257 a.a.) were generous gifts from Dr. K.L. Guan [45]. To generate pGEX4T-2-FOXO1-N terminal (1–157 a.a.) and pGEX4T-1-FOXO1-FK domain (158–258 a.a.) plasmids, the FOXO1 N terminal and FK domains were excised by *NcoI* and *MluI* digestion from the pCDNA3-FOXO1-N (1–257 a.a.) plasmid, then blunted by Klenow DNA polymerase and inserted in-frame into the *SmaI* site of pGEX4T-2 and pGEX4T-1 (Amersham Pharmacia Biotech), respectively. The FOXO1-M domain was released by digesting pCDNA3-Flag-FOXO1-C (259–416) with *EcoRI* and *NcoI*, blunting by Klenow DNA polymerase and inserting in-frame into the *SmaI* site of pGEX4T-1 to generate a pGEX4T-1-FOXO1-M plasmid. To generate the pGEX4T-2/FOXO1-C terminal (417–655 a.a.) plasmid, the C-terminal was released from the pCMV6-XL4/FOXO1-C terminal (417–655 a.a.) plasmid (which was generated by digestion of pCMV6-XL4/FOXO1 with *EcoRI* and then re-ligation) and inserted in-frame into pGEX4T-2. 1.4 kb-*SFTPB*-Luc containing a 1.4 kb fragment of the human surfactant protein B (*SFTPB*) promoter region that was cloned into the pGL2 basic vector [74]. pGL4.10-*TG(A)*, containing 2.5 kb of the human thyroglobulin (TG) promoter cloned into the *KpnI/XhoI* sites of the pGL4.10 vector (Promega, Madison, WI, USA) was a gift from Dr. Mihaela Stefan (Mount Sinai Medical Center, New York, USA) [75].

### 2.4. Antibodies and Reagents

The antibodies that were used for Western blotting analysis and immunostaining were as follows: FOXO1 (#2880 or #2880S, Cell Signaling, Danvers, MA, USA, 1:500); FOXO3 (#9467, Cell Signaling); FOXO4 (#9472, Cell Signaling, 1:500); p-FOXO1 (#ANTY011115, Antagene, Sunnyvale, CA, USA, 1:500); pro-SP-C (#AB3786, Millipore, Billerica, MA, USA, 1:500); AKT (#9272, Cell Signaling, 1:1000); p-AKT (ser473) (#3787, Cell Signaling, 1:1000); Lamin A/C (#SC20681, Santa Cruz Biotechnology, Santa Cruz, CA, USA, 1:1000); NKX2.1 (#MS-699-P0, Thermo Scientific, Pittsburgh, PA, USA, 1:500); GAPDH (#AM4300, Applied Biosystems, Austin, TX, USA, 1:1000), EIF-4E (#610270, BD Biosciences, San Jose, CA, USA, 1:200); eIF2*α* (#11386, Santa Cruz Biotechnology, 1:500); *β*-ACTIN (#ab6276, Abcam, Cambridge, MA, USA, 1:2000); and P180 lamellar body protein (#MMS-645R, Covance, Princeton, NJ, USA, 1:2000). VIIIB2 is a monoclonal antibody (mAb) that is specific for rat AT1 cells in situ that was previously generated in our laboratory (1:500) [76]. KGF, Ly294002 (PI-3K/AKT inhibitor) and isopropyl *β*-D-1-thiogalactopyranoside (IPTG) were purchased from R&D (Minneapolis, MN, USA), EMD Biosciences (San Diego, CA, USA) and Sigma, respectively.

### 2.5. Western Blotting Analysis 

Total protein was lysed in SDS sample buffer (2% SDS, 10% glycerol, 5% *β*-mercaptoethanol, pH 6.8). The samples were resolved by SDS-PAGE and electrophoretically blotted onto Immuno-Blot polyvinylidene fluoride (PVDF) membranes (Bio-RAD, Hercules, CA, USA). The membranes were blocked in either 5% nonfat dry milk or 3% BSA followed by incubation with the corresponding primary Abs at 4 °C overnight. After washing with TBS-T (20 mM Tris-7.5, 0.5 M NaCl, 0.01% Tween-20), the blots were incubated with horseradish peroxidase-linked anti-IgG conjugates (Promega) for 45 min at room temperature (RT). The complexes were visualized by enhanced chemiluminescence (ECL) (Pierce, Rockford, IL, USA) with an Alpha Ease Imaging System (Alpha Innotech, San Leandro, CA, USA). Lamin A/C, GAPDH, *β*-ACTIN, and EIF2*α* were used as controls for protein loading.

### 2.6. Nuclear Fractionation and Co-Immunoprecipitation (Co-IP)

Nuclear extraction and co-IP were performed by following the instructions for the Nuclear Complex Co-IP kit (Active Motif, Carlsbad, CA, USA). Briefly, Protein A/G plus agarose beads (Santa Cruz Biotechnology) were prewashed with 750 µL IP Low Wash Buffer (Active Motif) three times and resuspended in 120 µL IP Low Wash Buffer. The cell lysates were then precleared with prewashed Protein A/G beads. The precleared nuclear extracts (200 µg) were incubated with 2 µg of anti-Nkx2.1 monoclonal antibody or rabbit polyclonal anti-FOXO1 antibody or IgG (control) in 500 µL of IP Low Wash Buffer overnight at 4 °C. The following day, 150 µL of the prewashed Protein A/G beads was added and incubated for an additional 4 h at 4 °C followed by centrifugation at 3000 rpm for 30 sec. The pelleted beads were washed and resuspended in 30 µL of 4× SDS loading buffer (240 mM Tris pH 6.8, 26% glycerol, 0.1% DTT, 8% SDS, and 0.06% bromophenol blue) followed by boiling for 5 min. The supernatants were collected and loaded onto 7.5% gels for electrophoresis and subsequently probed for either FOXO1 (1:500) or NKX2.1 (1:500).

### 2.7. Purification of GST Fusion Proteins

*Escherichia Coli (E. Coli)* BL21 bacteria harboring pGEX-2T-NKX2.1, pGEX-2T-NKX2.1-N-terminal (1–371 a.a.), pGEX-2T-NKX2.1-homeodomain (HD) (141–253 a.a.) and pGEX-2T-Nkx2.1-C-terminal (254–371 a.a.), pGEX-KG-FOXO1, pGEX-4T-2-FOXO1-N terminal (1–157 a.a.), pGEX-4T-2-FOXO1-N+FK domain (1–257 a.a.), pGEX-4T-1-FOXO1-FK domain (158–258 a.a.), pGEX-4T-1-FOXO1-M terminal (259–416 a.a.), or pGEX-4T-1-FOXO1-C terminal (417–655 a.a.) domains were grown in 15 mL of Luria-Bertani (LB) media overnight. The cultures were then transferred to 250 mL LB medium and grown to an optical density of 0.7–0.8 followed by the addition of IPTG (1 mM final) (Sigma) and further cultured for 2 h at 28 °C to induce fusion protein expression. Bacteria were then pelleted, resuspended in 35 mL PBS, and sonicated (2 min × 2), followed by the addition of 1.5 mL 20% Triton-X 100 and incubated for 30 min at 4 °C. After centrifugation for 10 min at 12,000× *g*, the supernatants were transferred to a new 50 mL tube followed by the addition of 50% glutathione-sepharose 4 Fast Flow Beads (Amersham Pharmacia Biotech, Piscataway NJ, USA) and rotated for 30 min at RT. The mixture was then transferred to Poly-Prep Chromatography Columns (Bio-Rad). The GST fusion proteins were eluted with elution buffer (10 mM glutathione in 50 mM Tris-HCl at pH 8.0) after washing the columns three times with 2 mL PBS and dialyzed in PBS at 4 °C overnight.

### 2.8. GST Pull-Down Assay

The NKX2.1 and FOXO1 proteins were synthesized by in vitro translation in the presence of methionine using TNT T7 Quik Coupled Transcription/Translation Reaction kit (Promega). Translated NKX2.1 and FOXO1 were precleared by incubation with glutathione-sepharose. To evaluate the interaction between NKX2.1 and the specific domains of FOXO1, in vitro translated NKX2.1 was incubated with an equal volume of GST-FOXO1 fusion protein (full-length and different domains)-adsorbed glutathione-sepharose in the same volume of binding buffer containing 50 mM Tris-HCl, 120 mM NaCl, 2 mM EDTA, and 0.1% NP-40 for 1 h at RT. After extensive washing, the adsorbed protein complexes were boiled and analyzed by Western blotting using an anti-NKX2.1 antibody. GST adsorbed glutathione-sepharose was used as control. To determine the interaction between FOXO1 and the specific domains of NKX2.1, in vitro translated FOXO1 was incubated with GST-NKX2.1 fusion protein (full-length and different domains)-adsorbed glutathione-sepharose in binding buffer. After extensive washing, the adsorbed protein complexes were boiled and analyzed by Western blotting using an anti-FOXO1 antibody at a dilution of 1:500.

### 2.9. Electrophoretic Mobility Shift Assay (EMSA)

A DNA probe (5′-TAGGCCAAGGGCCTTGGGGCTCT-3′) containing the NKX2.1 binding site of the mouse *Sftpc* promoter (−186/−163) [77] was labeled using a biotin 3′-end labeling kit according to the manufacturer’s instructions (Pierce). Nuclear extracts (~4 μg) from MLE-15 cells were isolated using a nuclear extraction kit (Panomics, Redwood City, CA, USA) and ~2 × 10^5^ cpm of biotin-labeled oligonucleotide was incubated in EMSA buffer (20 mM Tris pH7.5, 2 mM NaCl, 2 mM EDTA, 10% glycerol, 2 mM DTT (freshly added), and 0.2 μg poly (dI/dC) (Amersham Pharmacia Biotech)) for 20 min at 4 °C, then incubated with increasing amounts of unlabeled purified glutathione-S-transferase (GST)-FOXO1, GST-FOXO1-FK or GST alone for 20 min. The DNA-protein complexes were then separated on a 7% acrylamide-GTG non-denaturing gel in 0.5× TBE (1× TBE: 89 mM Tris, 49 mM boric acid, 2 mM EDTA) and transferred to nylon membranes for detection using the Light-shift EMSA kit (Pierce).

### 2.10. Transfections

Transient transfections of MLE-15 cells were performed using Lipofectamine^™^ 2000 (Invitrogen, Carlsbad, CA, USA). MLE-15 cells were seeded at 5 × 10^4^ cells/well in a 24-well plate one day prior to transfection. The following day, the cells were transfected with 0.6 µg of *SFTPC* reporter (either 3.7-*SFTPC*-Luc or p318mu-*Sftpc*-Luc), 25 ng of pRC/CMV/*NKX2.1*, and 250 ng of pCDNA3/*FOXO1*, pCDNA3/*FOXO1-AAA* or pCDNA3/H215R plasmids. To examine the effects of FOXO1 on *SFTPB* promoter activity, 0.6 μg of *SFTPB* reporter (1.4 kb-*SFTPB*-Luc), 25 ng of pRC/CMV/*NKX2.1*, and 250 ng of pCDNA3/*FOXO1* were co-transfected into MLE-15 cells. To examine the effect of FOXO1 on thyroglubulin (*Tg*) reporter activity, 0.6 μg of pGL4.10-*TG(A*)-Luc, 50 ng of pRC/CMV/*NKX2.1*, and 50 ng of pCDNA3/*FOXO1* were transfected. 48 hours later, the cells were harvested, and luciferase activity was determined with the dual-luciferase reporter assay (Promega). Firefly luciferase was normalized to Renilla luciferase activity. 

### 2.11. Immunofluorescence Microscopy of Primary AEC 

Primary AEC monolayers that were grown on polycarbonate filters or cytospin slides of freshly isolated crude AEC were fixed with 4% paraformaldehyde (PFA) in phosphate buffered saline (PBS; pH 7.4) at RT or cold methanol, respectively, for 10 min. The filters were boiled with antigen-retrieval reagent (Life Technologies, Carlsbad, CA, USA) and blocked for 1 h with CAS block (Life Technologies) at RT, followed by incubation with p-FOXO1 antibody (1:300) overnight at 4 °C and biotinylated anti-rabbit IgG (1:250) (Vector). For p-FOXO1/P180 lamellar body protein or p-FOXO1/VIIIB2 double staining, cells on cytospin slides were incubated with rabbit anti-p-FOXO1 (1:1000) and either mouse anti-lamellar (1:2000) (Covance, Princeton, NJ, USA) or VIIIB2 (1:500) Abs overnight at 4 °C, followed by biotinylated anti-rabbit IgG (1:250) (Vector) and Alexa 594 anti-mouse IgG (1:1000) (Vector) for 1 h, respectively, and then avidin-conjugated fluorescein isothiocyanate (FITC) (1:250) (Vector) for 5 min. The cells on the filters and slides were mounted in Vectashield antifade mounting medium (Vector) with 4′,6-diamidino-2-phenylindole (DAPI) or propidium iodide (PI) for nuclear staining. The slides were viewed with an Olympus BX60 microscope that was equipped with epifluorescence optics (Olympus, Melville, NY, USA) and the images were captured using monochrome filters for fluorescein isothiocyanate (FITC) or rhodamine isothiocyanate (RITC) with a cooled charge-coupled device camera (Olympus, Melville, NY, USA).

### 2.12. Preparation of Lentivirus Expressing FoxO1 shRNA 

Human 293T cells at 80% to 90% confluence (ATCC, Manassas, VA, USA) were co-transfected by calcium phosphate precipitation with 12 μg of plasmids expressing *FoxO1* shRNA (clone ID TRCN0000054879 or TRCN0000054880 from Thermo Open Biosystems, Lafayette, CO, USA) or control non-silencing shRNA (clone ID SHC002, Sigma-Aldrich), 10 μg of pCMVDR8.91 for viral packaging, and 8 μg of pMD.G for VSV-G pseudotyping. The virus-containing supernatant from the transfected cells was harvested 48 and 72 h following transduction. The supernatant was filtered with 0.45 µm filters and centrifuged at 13,000× *g*. The pellets were resuspended in DMEM-F12 medium. Titers of virus stocks were determined by p24 Elisa Assay Kit (Cell Biolabs, San Diego, CA, USA).

### 2.13. Lentiviral Transduction of AEC and MLE-15 Cells

AEC were transduced on day 2 by adding 0.2 mL fresh medium containing lentivirus-expressing *FoxO1* shRNA or non-silencing shRNA at a multiplicity of infection (MOI) of 10 in the presence of polybrene (8 µg/mL) and the cell lysates were harvested for Western blotting at day 8. The MLE-15 cells that were grown on 24 well plates were transduced at MOI of 4 and cells were harvested 4 days post-transduction. 

### 2.14. High Dimensional Data Analysis

The publicly available ChIP-seq on NKX2-1 binding from *Sftpc^+^* AT2 (Rep 1: GSM4795154, Rep2: GSM4795155) and *Wnt3a^+^* AT1 cells (Rep1: GSM47951551, Rep2: GSM7955150) were downloaded from the Gene Expression Omnibus (GSE158196). Bedgraph files aligned to the mm10 genome were converted to bigwig using the UCSC wig_to bigWig tool (ucsc-wigtobigwig v357 hosted by Galaxy portal (https://usegalaxy.org/), accessed on 28 December 2021.) prior to visualization.

### 2.15. Statistical analysis

The data are shown as the mean ± standard error of the mean (SEM), where (*n*) is the number of observations. Significance (*p* < 0.05) was determined by two-tailed *t*-tests for comparison of two group means, and one-way ANOVA (with Turkey’s multiple comparisons test) for three or more group means. Statistical testing was performed using Prism 8 (GraphPad, v8. https://www.graphpad.com/ (accessed on 19 November 2020)).

## 3. Results 

### 3.1. FOXO1 Represses Expression of AT2 Cell-Specific Marker SP-C 

Using our in vitro AEC culture model in which AT2 cells transdifferentiate to an AT1 cell-like phenotype over a period of eight days, we examined the expression of FOXO isoforms during AT2 to AT1 differentiation to determine which FOXO family members may be involved in AEC differentiation. Western blotting analysis of protein that was harvested from freshly isolated rat AT2 cells (D0) and differentiating cells on days (D) 1, 3, 5, and 8 in culture showed that the FOXO1 protein levels were nearly undetectable on D0 and increased steadily and significantly from D1 to D8 (AT1-like cells) in culture (Figure 1A,B). The expression levels of the other two FOXO family members, FOXO3 and FOXO4, did not change significantly over time in culture (Figure 1A,C,D), although FOXO3 showed a trend toward an increase in the early days in culture (D1 and D3). We next examined the effects of FOXO1 knockdown on the expression of the AT2 cell-specific marker pro-SFTPC. As shown in Figure 1E–G, the knockdown of FOXO1 in primary rat AEC increased the expression of pro-SFTPC protein compared to the non-silencing shRNA. Similarly, pro-SFTPC levels increased following *FoxO1* knockdown in a mouse epithelial cell line, MLE-15 cells (Appendix A). These data suggest that FOXO1 regulates AEC phenotype and negatively regulates the expression of the AT2 cell marker SFTPC.

### 3.2. FOXO1 Inhibits NKX2.1-Induced Human SFTPC Promoter Activity

*SFTPC* is known to be transcriptionally activated by NKX2.1, a homeodomain-containing TF that binds to a proximal NKX2.1 binding site in the *SFTPC* promoter [77]. To examine potential effects of FOXO1 on NKX2.1-mediated *SFTPC* expression, MLE-15 cells were co-transfected with a luciferase reporter containing the 3.7 kb human *SFTPC* promoter (3.7-*SFTPC*-Luc) together with a NKX2.1 expression plasmid (pRC/CMV/*NKX2.1*) and/or a FOXO1 expression plasmid (pCDNA3/*FOXO1*). Co-transfection of NKX2.1 activated 3.7-*SFTPC*-Luc by ~9-fold, while the addition of FOXO1 significantly reduced NKX2.1-induced *SFTPC* activation in a dose-dependent manner (Figure 2A,B). Since the knockdown of *FoxO1* did not alter NKX2.1 expression levels (Appendix A), these results suggest that FOXO1 inhibits SFTPC expression at least in part via NKX2.1-mediated *SFTPC* transcriptional activity, although we cannot exclude the inhibition of translational activity. We next examined whether FOXO1 repression of NKX2.1-activated gene expression could be generalized to other NKX2.1-regulated genes. FOXO1 exerted similar repressive effects on NKX2.1-mediated *SFTPB* promoter activity (Figure 2C) but had no effect on thyroglobulin-luciferase promoter activity (Appendix A). These results suggest that FOXO1 repression of target genes is AT2 cell-specific.

### 3.3. FOXO1 DNA Binding Ability Is Not Required for Repression of NKX2.1-Mediated SFTPC Expression 

FOXO1 can regulate target gene expression by either directly binding to DNA or in a DNA-independent manner through the interaction with other TFs [46]. The *SFTPC* promoter contains a FOX consensus binding site at −1722/−1716 (Figure 2D) that is conserved among rats, mice, and humans [73]. To investigate whether the FOXO1 repressor function is mediated through direct DNA binding to the *SFTPC* promoter, we performed transient transfections using a FOXO1 mutant (H215R) expression construct that bears a mutation in the DNA binding domain and thus cannot bind to DNA [45]. MLE-15 cells were co-transfected with 3.7-*SFTPC*-Luc and NKX2.1 and FOXO1 H215R expression plasmids. FOXO1 H215R repressed the NKX2.1-mediated activation of *SFTPC* promoter transcription similarly to wild-type (WT) FOXO1 (Figure 2E). Additionally, FOXO1 repressed the NKX2.1-mediated activation of mouse p318mu*Sftpc*-Luc, a shorter promoter construct that lacks the Fox DNA binding site, but contains NKX2.1 binding sites that are conserved with human *SFTPC* [78] in a dose-dependent manner (Figure 2F). These data demonstrate that FOXO1-mediated attenuation of *SFTPC* induction by NKX2.1 is independent of direct binding of FOXO1 to its cognate binding site in the *SFTPC* promoter and likely involves protein–protein interactions.

### 3.4. FOXO1 and NKX2.1 Interact during AEC Differentiation

Since FOXO1 knockdown did not affect NKX2.1 expression (Appendix A) and the inhibition of NKX2.1-mediated transcriptional activation of *SFTPC* did not require the DNA binding domain of FOXO1, we investigated whether interactions between FOXO1 and NKX2.1 might be driving reductions in SFTPC levels during differentiation. We performed co-IP assays using cell lysates that were harvested on D3 from cultured primary rat AEC, a transitional time-point at which SFTPC expression was first absent, while FOXO1 expression was starting to increase (Figure 1A,B) and NKX2.1 was still present [73]. Immunoprecipitation of the extracts with an anti-FOXO1 antibody followed by immunoblotting for NKX2.1 demonstrated that NKX2.1 interacts with FOXO1 (Figure 3A). Immunoprecipitation of NKX2.1 followed by immunoblotting for FOXO1 similarly confirmed the association between FOXO1 and NKX2.1 (Figure 3B). These data provide strong evidence that NKX2.1 and FOXO1 proteins interact during AEC differentiation.

### 3.5. Forkhead Domain of FOXO1 Physically Interacts with the Homeodomain of NKX2.1

To elucidate the mechanisms underlying FOXO1-mediated inhibition of *SFTPC* expression, we next sought to determine how FOXO1 and NKX2.1 physically interact. FOXO1 protein contains an N-terminal repression domain (N), a forkhead domain harboring the DNA-binding domain (FK), a middle region possessing a nuclear export signal (M), and a C-terminal domain that includes a transactivation domain (C) [79]. We generated a series of FOXO1 GST fusion constructs harboring these different domains (left panel in Figure 3C and Appendix A). The incubation of in vitro translated full-length NKX2.1 with immobilized GST-FOXO1 (full-length (FL)) or GST-FOXO1-N, GST-FOXO1-FK, GST-FOXO1-N+FK, GST-FOXO1-M, and GST-FOXO1-C domain fusion proteins demonstrated that NKX2.1 interacts with full-length FOXO1 as well as with FK domains (lane 2, 4, and 5, right panel in Figure 3C). NKX2.1 did not interact with the N-terminal, M, or C-terminal domains (lane 3, 6, and 7, right panel in Figure 3C), indicating a specific role for the FK domain in NKX2.1 interaction. GST alone, a negative control, did not interact with NKX2.1 (lane 1, right panel in Figure 3C). To identify the interacting domain of NKX2.1, we performed GST pull-down assays by incubating in vitro translated full-length FOXO1 with immobilized GST-NKX2.1 (FL) (containing full-length NKX2.1), GST-NKX2.1-N (containing N-terminal NKX2.1), GST-NKX2.1-HD (containing homeodomain (HD) NKX2.1), or GST-NKX2.1-C (containing C-terminal NKX2.1) domain fusion proteins (left right panels in Figure 3D and Appendix A). The homeodomain of NKX2.1 interacted directly with FOXO1, while the N and C domains did not (right panel in Figure 3D). These results demonstrate a direct physical interaction between the FK domain of FOXO1 and the homeodomain of NKX2.1.

### 3.6. FOXO1 Binding to NKX2.1 Interferes with Binding of NKX2.1 to the Sftpc Promoter

To address whether the interaction of the FOXO1 FK domain with the NKX2.1 homeodomain interferes with the DNA binding ability of NKX2.1 at the *Sftpc* promoter, we performed EMSAs using nuclear extracts that were harvested from MLE-15 cells that expressed high endogenous levels of NKX2.1. As previously reported [73], the nuclear extracts from the MLE-15 cells form a complex with the −186/−163 bp mouse *Sftpc* promoter probe containing a consensus NKX2.1 binding motif (lane 2) (Figure 3E,F). Recombinant GST-FOXO1 fusion protein (Figure 3E) or GST-FOXO1 FK domain fusion protein (Figure 3F) inhibited the formation of NKX2.1-DNA complexes in a dose-dependent manner (lanes 8 to 10). Equimolar GST alone had no effect on the formation of NKX2.1/probe complexes (lanes 4 to 6), indicating that the inhibitory effect of FOXO1 on the DNA binding capacity of NKX2.1 to the *Sftpc* promoter is specific. Analysis of publicly available data on the binding of NKX2-1 in mouse *Sftpc^+^* AT2 and *Wnt3a^+^* AT1 cells [31] shows that binding of NKX2-1 to sites near AT2 cell-specific genes *Sftpc* and *Sftpb*, in the vicinity of the predicted NKX2.1 binding sites (−171 to −181 for *SFTPC* and −123 to −113 for *SFTPB* from the transcriptional initiation sites), decreased concomitantly with a known loss of surfactant expression during AT2-AT1 differentiation (Figure 3G). This decrease was not due to an overall decreased binding of NKX2-1 in the AT1 cells as increased NKX2.1 binding at numerous genes, including known AT1 cell markers, as well no change in NKX2-1 binding dynamics at several sites near known AT1 genes (Appendix A) were observed. Together these findings support the notion that FOXO1 interaction with the NKX2.1 homeodomain interferes with its ability to bind to and transcriptionally activate the *SFTPC* promoter.

### 3.7. PI-3K/AKT-Mediated FOXO1 Phosphorylation Regulates FOXO1 Repression of NKX2.1-Mediated Transcriptional Activation of SFTPC

We next sought to investigate the mechanisms that could regulate FOXO1 interactions with NKX2.1. FOXO1 nuclear translocation and activity are regulated by phosphorylation/dephosphorylation, with unphosphorylated FOXO1 being nuclear and active while phosphorylated FOXO1 is cytoplasmic and becomes degraded by the ubiquitination proteosome pathway [74]. To investigate whether FOXO1 phosphorylation/dephosphorylation regulates its effects on NKX2.1-mediated *SFTPC* activation, MLE-15 cells were transfected with the *SFTPC* promoter reporter 3.7-SFTPC-Luc, pRC/CMV/*NKX2.1*, as well as either pCDNA3 *FOXO1* or a constitutively active form of FOXO1 (pCDNA3 *FOXO1-AAA*). Compared to the wild-type FOXO1, co-transfection with the FOXO1-AAA mutant showed a trend toward further inhibiting NKX2.1-mediated transactivation of the *SFTPC* promoter (Figure 4A) suggesting FOXO1 phosphorylation decreases its inhibitory effects on NKX2.1-mediated activation of *SFTPC*. The addition of the PI-3K/AKT inhibitor Ly294002 to co-transfections in the MLE15 cells with the 3.7-kb human *SFTPC* promoter and NKX2.1 expression plasmid decreased p-FOXO1 and p-AKT levels and attenuated NKX2.1-mediated transactivation of the *SFTPC* promoter activity by ~60% (Figure 4B,C), further supporting that non-phosphorylated FOXO1 is necessary for NKX2.1-mediated loss of surfactant gene activation. To determine whether FOXO1 phosphorylation affects its binding to NKX2.1, co-IP was performed using nuclear extracts that were harvested from MLE-15 cells that were treated with Ly294002 (6 μM) for 48 h. Ly294002 treatment increased the association between FOXO1 and NKX2.1 (Figure 4D) suggesting that PI-3K/AKT negatively regulates the association between FOXO1 and NKX2.1.

### 3.8. FOXO1 Activity Is Modulated by Phosphorylation during Alveolar Epithelial Cell Differentiation

Given that FOXO1 interaction with NKX2.1 disrupts NKX2.1-activated *SFTPC* expression and that phosphorylation alters the interaction of FOXO1 and NKX2.1, we examined the dynamic changes in FOXO1 phosphorylation/dephosphorylation in the context of rat AEC differentiation. We performed co-immunostaining of freshly isolated rat AEC (containing both AT2 and AT1 cells) with anti-phosphorylated FOXO1 (p-FOXO1)/anti-P180 (an AT2 cell-specific marker), or with anti-p-FOXO1/antiVIIIB2 (an AT1 cell-specific marker) antibodies. p-FOXO1 colocalized with lamellar body protein but not with VIIIB2 (Figure 5A). Together with Figure 1A, these results suggest that the phosphorylated form of FOXO1 predominates in AT2 cells, and that unphosphorylated FOXO1 is the predominant form that is found in AT1 cells. We previously reported that treatment of AEC in vitro with KGF maintains the AT2 cell-like phenotype [6]. Consistent with the notion that FOXO1 is phosphorylated in AT2 cells, KGF treatment from D0 to D8 increased p-FOXO1 and the expression of the AT2 cell marker pro-SFTPC. A similar trend was observed with KGF treatment of AT1.5 intermediate cells (D4) through D8 of AEC differentiation (Figure 5B–D). We next examined whether phosphorylation of FOXO1 was regulated by the PI-3K/AKT pathway during AEC differentiation. The treatment of AEC with the PI-3K/AKT inhibitor Ly294002 from D4–D8 decreased KGF-induced p-FOXO1 and pro-SFTPC expression (Figure 5E–H). Furthermore, the treatment of primary rat AEC with KGF resulted in an increase in phosphorylated AKT within 15 min which was sustained up to 6 h, while the treatment of the cells with Ly294002 markedly reduced p-AKT levels (Figure 5I,J). These data suggest that KGF plays an important role in regulating AT2 cell-specific SFTPC expression and maintaining AT2 cell phenotype via PI-3K/AKT-dependent phosphorylation of FOXO1.

## 4. Discussion 

FOXO1 has been implicated in diverse biological processes ranging from cell proliferation, differentiation, and apoptosis in various tissues to tissue-specific gene expression; however, the function of FOXO1 in the regulation of AEC phenotype in the distal lung has not been previously investigated. Here, we report that FOXO1 protein–protein interaction with NKX2.1 during AEC differentiation in the adult lung results in loss of surfactant gene expression [31,63,64,70]. Specifically, the FK domain of FOXO1 interacts with the homeodomain of NKX2.1 to disrupt binding of NKX2.1 to the *SFTPC* promoter. The inhibitory effect of FOXO1 is regulated by PI-3K/AKT signaling. Our in vitro primary culture model further demonstrates a role for FOXO1 in AT2 progenitor cell maintenance and differentiation as well as the regulation of AT2 cell-specific gene expression via coupling of KGF and the PI-3K/AKT pathways. Recently, three-dimensional (3-D) organoid cultures of mouse AT2 cells with fibroblasts have been used to study the effects of potential niche factors in regulating AT2 progenitor functions [4,80,81]. The role of KGF and PI-3K/AKT/FOXO1 signaling in AT2 cell proliferation and differentiation requires further study using a fibroblast-free 3-D culture system under conditions in which the AT2 cells both proliferate and differentiate into AT1 cells using small molecule inhibitor (e.g., Ly294002) and shRNA approaches. Additionally, the translation of our findings from rodents to human lung will require further investigation.

FOXO family members regulate key aspects of cell physiology. These include cell cycle progression, proliferation, differentiation, survival, longevity, and response to stress in mature cell types [39,82,83,84]. FOXO proteins seem overlapping, but not redundant [85,86]. Increasing evidence supports their key roles in maintenance and differentiation of tissue-specific stem/progenitor cells [84,87,88]. For example, FOXO3 was shown to be critical in the maintenance of neural stem cells (NSCs) as its inactivation leads to decreased self-renewal and an impaired ability of NSCs to generate different neural lineages [89]. FOXO4 is necessary for the differentiation of human embryonic stem cells (hESCs) into neural cells, while the loss of FOXO4 reduces the potential of hESCs to differentiate into neural lineages [90]. Both FOXO3 and FOXO4 were shown to be involved in maintaining the self-renewal capacity of HSCs [91,92,93] and cancer stem cells [93,94,95]. Similarly, FOXO1 is critical for both the maintenance and differentiation of stem/progenitor cells such as hESCs, spermatogonial stem cells, and mesenchymal stem cells [50,51,60]. 

In the present study, we found that FOXO1 is phosphorylated (i.e., inactive) in AT2 progenitor cells and FOXO1 activity is required for reduction of the AT2 cell marker SFTPC during differentiation into AT1-like cells, suggesting that active FOXO1 is not required for AT2 progenitor cell maintenance, but rather for AT2 to AT1 cell differentiation. Whereas FOXO1, FOXO3, and FOXO4 are all expressed in the lung and AEC, we show that expression of FOXO1, but not FOXO3 and FOXO4, increases significantly during AT2 to AT1-like differentiation, supporting a requirement for FOXO1 in this process. In addition, we found that FOXO1 represses the NKX2.1-activated expression of another AT2 cell-specific gene *SFTPB*, but not NKX2.1-activated thyroid-specific gene thyroglobulin (*Tg*) transcription. Thus, FOXO1 function is both tissue- and cell context-dependent. However, potential compensatory effects of FOXO3 and FOXO4 cannot be excluded. The trend towards FOXO3 upregulation at the early stage of AT2-AT1 cell differentiation (day one and three) and stabilization at the late stages of differentiation (day five and day eight) suggests FOXO3 may also be involved in early AEC differentiation, although this did not reach statistical significance. It will be interesting to assess double and/or triple FOXO1/FOXO3/FOXO4 knockout mice to study the specific role of FOXO1 and the redundancy of FOXO family members (e.g., FOXO3) during AT2 to AT1 cell differentiation.

NKX2.1 is known to regulate AT2 cell-specific genes, and its expression has also been recently reported in AT1 cells [31,65,70]. NKX2.1 regulates AT1 and AT2 cell-specific genes via both cell type-specific DNA binding as well as binding to sites that are common to both AT2 and AT1 cells in coordination with differential cell-specific interactions with other TFs. For example, a recent study identified that AT2 cell-specific NKX2.1 binding is dependent on CEBPs while AT1 cell-specific NKX2.1 binding is dependent on TEADs. YAP/TAZ/TEAD could direct NKX2.1 to AT1 cell-specific genes, while the loss of YAP/TAZ shifted NKX2.1 binding to AT2 cell-specific genes leading to AT2 cell gene expression [31,70,96]. Here we identify FOXO1 as a novel TF that modulates NKX2.1 DNA binding and activation of surfactant proteins via the interaction with its homeodomain, thus providing new insights into NKX2.1-mediated AT2- and AT1-cell fate determination. Additionally, we have previously shown that NKX2.1 activation of surfactant gene expression can be antagonized by other FOX family proteins, FOXP2 [73] and FOXA1 [97], suggesting that this competitive mechanism is more generally applicable to the regulation of AEC differentiation and further establishing the importance of FOX family members in the control of cell-specific gene expression during this process.

It is well-established that phosphorylation of FOXO1 at T24, S256, and S319 by insulin-activated AKT leads to its export from the nucleus to the cytoplasm with subsequent ubiquitination and degradation [98]. Our results show that FOXO1 expression is increased in AT1-like cells while p-FOXO1 is present primarily in the AT2 cells and KGF-maintained AT2 cells in primary culture. Furthermore, the PI-3K inhibitor Ly294002 decreases FOXO1 phosphorylation and increases the expression of SFTPC, suggesting that PI-3K/AKT-dependent FOXO1 inactivation plays a role in the regulation of *Sftpc*, consistent with insulin effects on FOXO1 in other tissues. However, FOXO1 is also known to be regulated by other kinases such as serum/glucocorticoid regulated kinase 1 (SGK1) [99,100] and whether other kinases mediate FOXO1 phosphorylation in AT2 cells in the absence of KGF would be interesting to explore. Additionally, co-IP demonstrated that Ly294002 increases the interaction of FOXO1 and NKX2.1, suggesting that FOXO1 activity may also be regulated by changes in binding affinity, with dephosphorylated FOXO1 having higher affinity for NKX2.1, leading to greater repression of SFTPC expression. KGF has been shown to protect the lung from a variety of insults [101,102]. Although several mechanisms including promotion of AT2 cell proliferation and activation of the pro-survival AKT pathway have been proposed to address the protective effect of KGF within the lung [103,104,105,106], mechanisms underlying KGF-mediated protection are not fully understood. Here we show that in KGF-treated primary AT2 culture, the expression of p-AKT, p-FOXO1, and the AT2 cell marker SFTPC increased. Together with the findings that both FOXO1 expression level and its binding to NKX2.1 are regulated by PI-3K/AKT signaling, our data suggest that KGF may play an important role in maintaining AT2 cell homeostasis via PI-3K/AKT-mediated downregulation of FOXO1 activity during repair following lung injury.

The lung alveolar epithelium is comprised of AT2 and AT1 cells. The AT2 cells serve as the primary progenitors for the AT1 cells in the adult lung, both self-renewing and differentiating into AT1 cells to restore normal lung function following various injuries. Although the mechanisms underlying AT2 cell proliferation have been more intensively studied, recent studies have identified several signaling pathways that are involved in the regulation of AT2 to AT1 cell differentiation [107]. For example, canonical Wnt signaling blocks the differentiation and abrogation of Wnt signaling (e.g., treatment with Wnt antagonist Dickkopf 3 or deletion of β-catenin) promotes it [19], while non-canonical WNT5a/protein kinase C signaling promotes AT2 cell differentiation [26]. Notch, BMP, and TGF-β signaling are temporally regulated as AT2 cells transition from proliferation to differentiation. Notch signaling is initially activated during proliferation but is downregulated via Dlk1 during subsequent AT1 cell differentiation [33]. The deactivation of BMP signaling promotes AT2 cell proliferation, but activation promotes AT2 to AT1 cell differentiation [34]. TGF-*β* signaling is initially low in proliferating AT2 cells, highly upregulated in the intermediate cell state, and subsequently downregulated in differentiating cells [36]. Furthermore, YAP/TAZ signaling is activated in both AT2 and AT1 cells, with AT2 cell-specific deletion of YAP or TAZ inhibiting AT2 cell proliferation as well as AT2 to AT1 cell differentiation during regeneration [21,22,23,30]. FGF signaling is essential for AT2 cell homeostasis and KGF and FGF10 stimulate AT2 cell proliferation, while FGFR2 loss decreased AT2 cell proliferation and increased differentiation into AT1 cells in damaged lungs [17,108,109]. We have previously reported that AEC that are treated with KGF from day four to day eight reacquire lamellar bodies and demonstrate partial reversion to a cuboidal AT2 cell-like morphology suggesting that indeed, this signaling network may be reversible [6]. In the current study, our results show that KGF and PI-3K/AKT regulate the maintenance of the AT2 cell phenotype and AT2 to AT1 cell differentiation through FOXO1, which can be regulated by Ly294002 a PI-3K/AKT pathway inhibitor. Together, these studies suggest complex networks that are regulating AT2 to AT1 cell differentiation [69] and a role for PI-3K/AKT/FOXO1 pathway in the reprogramming of AT1 cell into AT2 cells following injury. Whether and how KGF and PI-3K/AKT/FOXO1 signaling interact with these other pathways to coordinately regulate the differentiation process will require further investigation.

In summary, FOXO1 interacts with NKX2.1 to inhibit NKX2.1-DNA binding and transcriptional stimulation of surfactant proteins. This activity of FOXO1 is regulated via KGF and PI-3K/AKT-mediated signaling (Figure 6). Our results provide new insights into the important role of FOXO1 in regulating transcription of AT2 cell-specific genes via protein–protein interaction with NKX2.1 during AEC differentiation. These results also link KGF and PI-3K/AKT/FOXO1 signaling to NKX2.1-mediated AT2 and AT1 cell fate decisions during lung injury and repair.

## Figures and Tables

**Figure 1 cells-11-01122-f001:**
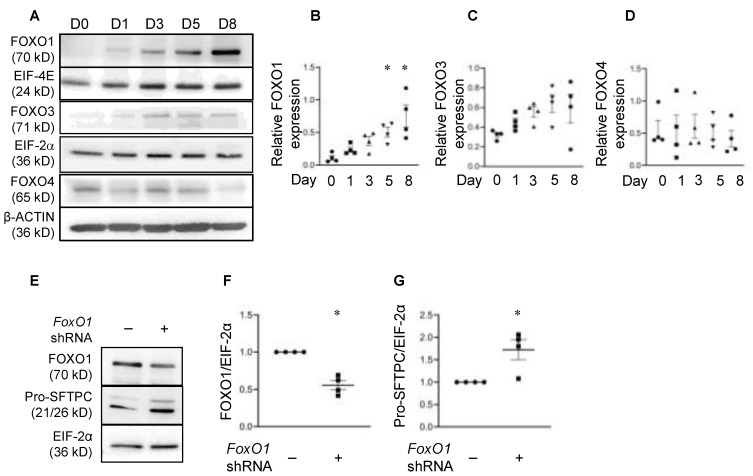
FOXO1 increases in alveolar epithelial cells as a function of time in culture, and knockdown increases expression of the AT2 cell-specific marker SFTPC. Representative Western blot (WB) (**A**) and quantitation (**B**–**D**) show that the expression of FOXO1, but not FOXO3 and FOXO4, increases as rat AT2 cells (D0) differentiate into AT1-like cells at day eight (D8) in primary culture. EIF-4E, EIF-2α, and *β*-ACTIN were the loading controls. *n* = 4 for each group. One-way ANOVA, asterisk indicates *p* < 0.05 compared to D0. Representative WB (**E**) and quantitation (**F**,**G**) show that knockdown of FOXO1 increased pro-SFTPC expression in primary AEC six days following transduction with *FoxO1* shRNA (+) or non-silencing shRNA (−) on Day two. EIF-2α was used as the loading control. The data were normalized to non-silencing shRNA. *n* = 4 for each group. Unpaired two-tailed *t*-test, asterisk indicates *p* < 0.05.

**Figure 2 cells-11-01122-f002:**
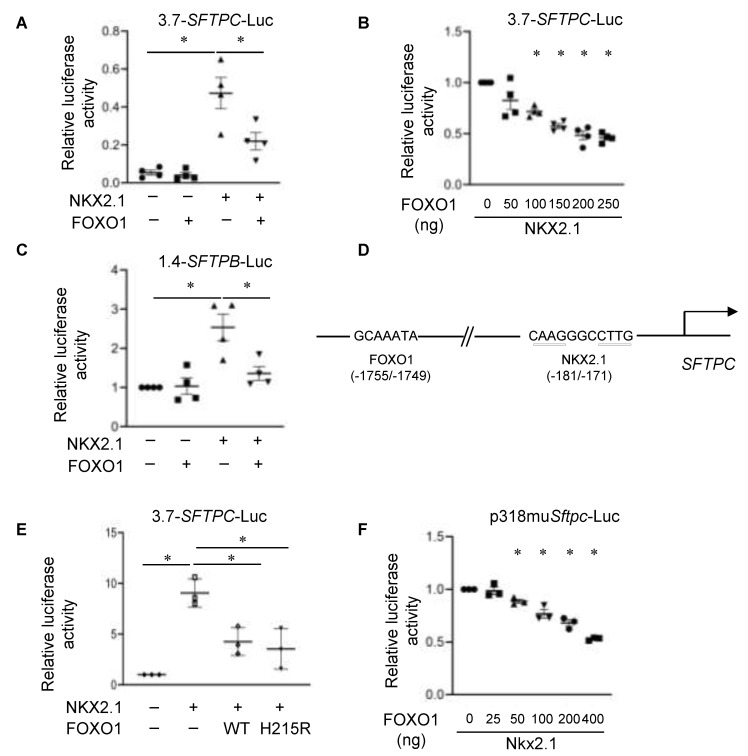
FOXO1 attenuates NKX2.1 activation of *SFTPC* and *SFTPB***.** (**A**) MLE-15 cells were co-transfected with a 3.7 kb *SFTPC* reporter construct (3.7-*SFTPC*-Luc) and NKX2.1 or FOXO1 expression constructs alone or in combination. Dual luciferase assays 48 h following transduction showed that FOXO1 inhibited NKX2.1 activation of the *SFTPC* reporter. Firefly luciferase activity was normalized to Renilla luciferase. *n* = 4 for each group. One-way ANOVA, asterisk indicates *p* < 0.05. (**B**) MLE-15 cells were co-transfected with 3.7-*SFTPC*-Luc, pRC/CMV/*NKX2.1* and increasing concentrations of FOXO1 expression construct. Dual luciferase assays 48 h post-transduction showed that FOXO1 inhibited NKX2.1-mediated induction of the *SFTPC* reporter in a dose-dependent manner. The data are shown as normalized to the absence of FOXO1. *n* = 4 for each group. One-way ANOVA, asterisk indicates *p* < 0.05 compared to absence of FOXO1 (empty vector). (**C**) MLE-15 cells were co-transfected with a 1.4 kb *SFTPB* reporter construct (1.4-*SFTPB*-Luc), NKX2.1 and FOXO1 expression constructs alone, or in combination. Dual luciferase assays 48 h post-transduction showed that FOXO1 inhibited NKX2.1 activation of the *SFTPB* reporter. Firefly luciferase activity was normalized to Renilla luciferase. The data are shown as normalized to the absence of FOXO1 and NKX2.1. *n* = 3 for each group. One-way ANOVA, asterisk indicates *p* < 0.05. (**D**) Schematic showing the putative FOXO1 and NKX2.1 binding sites on human *SFTPC* promoter. (**E**) MLE-15 cells were co-transfected with a 3.7 kb *SFTPC* reporter construct (3.7-*SFTPC*C-Luc), NKX2.1 and FOXO1 wild-type (WT), or FOXO1H215R (DNA binding mutant) expression constructs alone or in combination. Dual luciferase assays 48 h post-transduction showed that FOXO1-H215R maintained the ability to repress NKX2.1 activation of the *SFTPC* reporter. Firefly luciferase activity was normalized to Renilla luciferase. The data are shown as normalized to the absence of FOXO1 and NKX2.1. *n* = 3 for each group. One-way ANOVA, asterisk indicates *p* < 0.05. (**F**) MLE-15 cells were co-transfected with a 318 bp *Sftpc* reporter p318mu*sftpc*-Luc which lacks a FOXO1 binding site and a NKX2.1 expression construct, together with increasing amounts of FOXO1 expression construct. Dual luciferase assays 48 h post-transduction showed that FOXO1 inhibited NKX2.1-induced activation of the 318 bp *Sftpc* reporter in a dose-dependent manner. The data are shown as normalized to the absence of FOXO1 (empty vector). *n* = 3 for each group. One-way ANOVA, asterisk indicates *p* < 0.05 compared to absence of FOXO1.

**Figure 3 cells-11-01122-f003:**
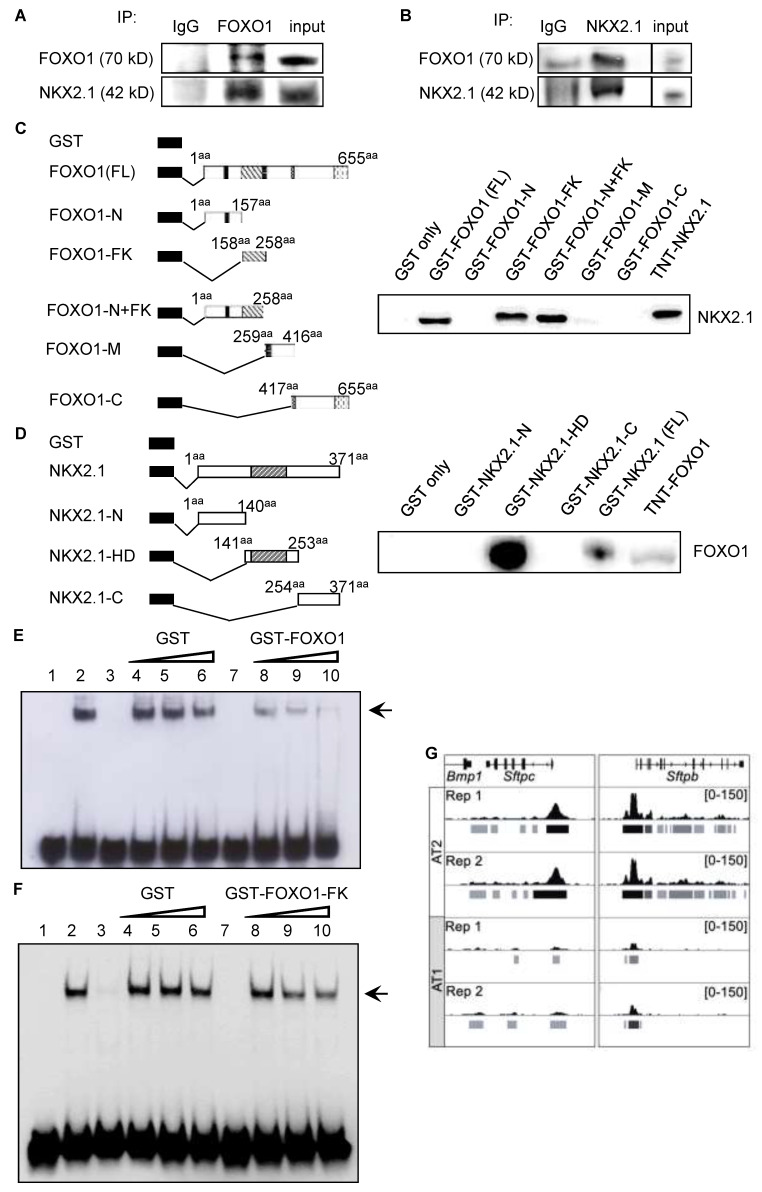
FOXO1 physically interacts with the homeodomain of NKX2.1. Co-immunoprecipitation of nuclear extracts that were harvested from primary AT2 cells on day three in culture with anti-NKX2.1 (**A**) or anti-FOXO1 (**B**) Abs shows the association of endogenous FOXO1 and NKX2.1 (*n* = 4). IgG was used as a negative control. (**C**) Schematic of GST-tagged FOXO1 fusion proteins (left panel) and GST pull-down assay (right panel). In vitro translated NKX2.1 was incubated with GST-FOXO1 fusion proteins (GST-FOXO1 full-length (FL), GST-FOXO1-N, GST-FOXO1-FK, GST-FOXO1-N + FK, GST-FOXO1-M, GST-FOXO1-C) coupled to glutathione sepharose. Bound NKX2.1 was visualized by Western blotting using an anti-NKX2.1 antibody. The FOXO1 FL and FOXO1 FK domains interact with NKX2.1 (right panel). *n* = 3. (**D**) Schematic of GST-tagged NKX2.1 fusion proteins (left panel) and GST pull-down assay (right panel). In vitro translated FOXO1 was incubated with GST- NKX2.1 fusion proteins (GST-NKX2.1 FL, GST-NKX2.1-N, GST-NKX2.1-HD, and GST-NKX2.1-C) that were coupled to glutathione sepharose. The bound FOXO1 was visualized by Western blotting using an anti-FOXO1 Ab. FOXO1 interacts with the NKX2.1 homeodomain (right panel). *n* = 3. (**E**,**F**) EMSA was performed with nuclear extracts from MLE-15 cells and biotin-labeled oligonucleotides encompassing the NKX2.1 DNA-binding site (−186 to −163 bp) of the *SFTPC* promoter. The arrow points to the inhibition of the NKX2.1 protein/DNA complexes (lane 2) with increasing amounts of GST-FOXO1 fusion protein (lane 8–10) (**E**) or GST-FOXO1 FK domain fusion protein (lane 8–10) (**F**) but not by GST alone (lane 4–6). Lane 1 is probe only. GST (lane 4) and GST-FOXO1 or GST-FOXO1 FK (lane 7) proteins do not form a complex with the oligonucleotide probe. *n* = 3. (**G**) Publicly available ChIPseq analysis of NKX2-1 binding surrounding the *Sftpc* and *Sftpb* loci in *Sftpc^+^* AT2 and *Wnt3A^+^* AT1 cells. Peaks were called using MACS2.0 as outlined by Little DR et al. [31]. 0–150 indicates number of Chip-Seq reads overlapping at a given base.

**Figure 4 cells-11-01122-f004:**
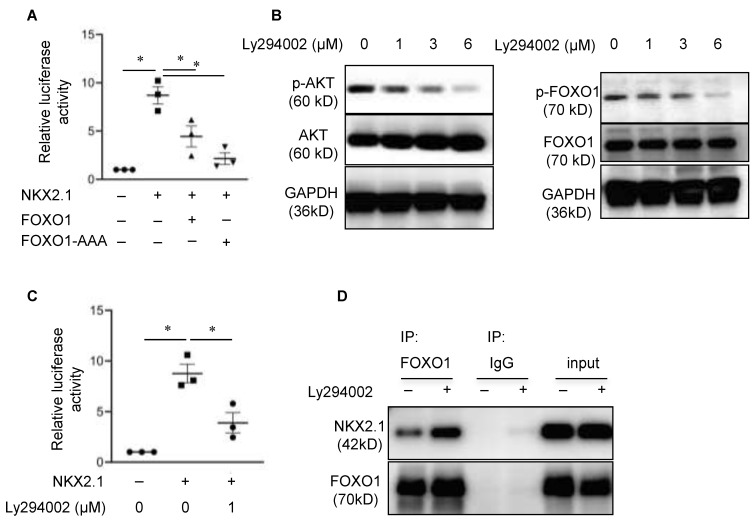
FOXO1 repressor function is negatively regulated by PI-3K-dependent phosphorylation. (**A**) MLE-15 cells were co-transfected with the 3.7-SFTPC-Luc reporter construct, NKX2.1 expression construct and FOXO1 or FOXO1-AAA, a dephosphorylated constitutively active form of FOXO1. Dual luciferase assays were performed 48 h after transfection. Firefly luciferase activity was normalized to Renilla luciferase. The data are shown as normalized to the absence of FOXO1 and NKX2.1. *n* = 3 for each group. One-way ANOVA, asterisk indicates *p* < 0.05. (**B**) Western blotting for *p*-AKT (ser473) and p-FOXO1 in MLE-15 cells shows that Ly294002 treatment (7 h) decreases p-AKT and p-FOXO1. *n* = 2. (**C**) MLE-15 cells were co-transfected with a 3.7-SFTPC-Luc reporter construct and NKX2.1 expression construct for 24 h, followed by treatment with Ly294002 (1 µM) for an additional 24 h. Dual luciferase assay showed that Ly294002 inhibited *SFTPC* reporter activity. The data are shown as normalized to the absence of both NKX2 and Ly294002. *n* = 3 for each group. One-way ANOVA, asterisk indicates *p* < 0.05. (**D**) Co-IP was performed with cell lysates that were harvested from MLE-15 cells that were cultured in the presence or absence of PI-3K inhibitor Ly294002 (6 μM) for 48 h. Increased association of NKX2.1 with FOXO1 was detected in the Ly294002-treated samples. *n* = 3.

**Figure 5 cells-11-01122-f005:**
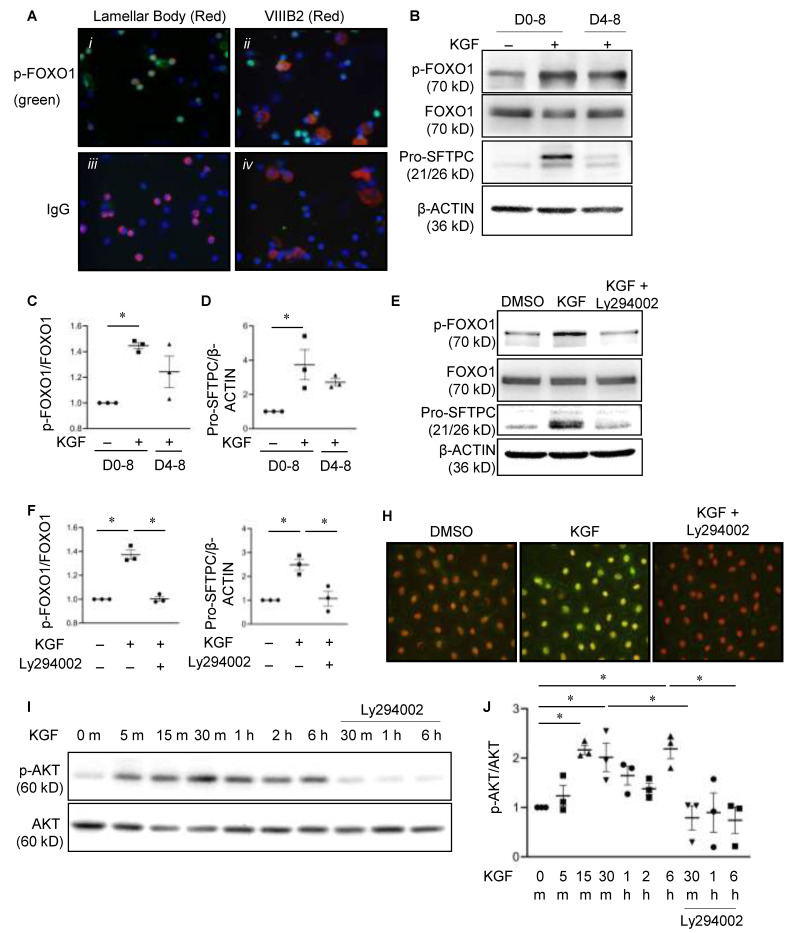
FOXO1 phosphorylation is regulated as a function of AEC differentiation by KGF in an AKT-dependent manner. (**A**) Double labeling immunofluorescence was performed on freshly isolated rat lung epithelial cells with p-FOXO1 (FITC, green) and either lamellar membrane protein P180 (LBM-180, an AT2 cell-specific marker) (red), (i) or VIIIB2 (an AT1 cell-specific marker) (red) (ii). The nuclei were stained with DAPI (blue). Species-specific IgGs were used as controls (iii and iv). p-FOXO1 co-localized with LBM-180, but not VIIIB2. *n* = 4. Representative Western blot (**B**) and quantitation of p-FOXO1 (**C**) and pro-SFTPC (**D**) in AT2 cells that were grown in primary culture from day zero (D0) to day eight (D8) and that were treated with KGF (10 ng/mL) from either D0–8 or D4–8 shows increased phosphorylation of FOXO1 and preserved expression of pro-SFTPC. *β*-ACTIN was the loading control. The data are shown as normalized to the absence of KGF. *n* = 3 for each group. One-way ANOVA, asterisk indicates *p* < 0.05. Representative Western blot (**E**) and quantitation of p-FOXO1 (**F**) and pro-SFTPC (**G**) in primary rat AEC that were treated with KGF ± Ly294002 from D4–D8. Ly294002 reduced p-FOXO1 and pro-SFTPC. *β*-ACTIN was a loading control. The data are shown as normalized to the absence of KGF and Ly294002. *n* = 3 for each group. One-way ANOVA, asterisk indicates *p* < 0.05. (**H**) Representative immunofluorescence analysis of p-FOXO1 (FITC, green) in AEC that were treated with KGF ± LY294002 from D4–8 shows that Ly294002 reduces p-FOXO1 expression. The untreated AEC (DMSO) were used as controls. The nuclei were counterstained with propidium iodide (PI). *n* = 4. Representative Western blot (**I**) and quantitation (**J**) of p-AKT (Ser473) in primary AT2 cells that were treated with KGF ± Ly294002. Ly294002 inhibits KGF-induced p-AKT expression. Total AKT was a loading control. *n* = 4 for each group. One-way ANOVA, asterisk indicates *p* < 0.05.

**Figure 6 cells-11-01122-f006:**
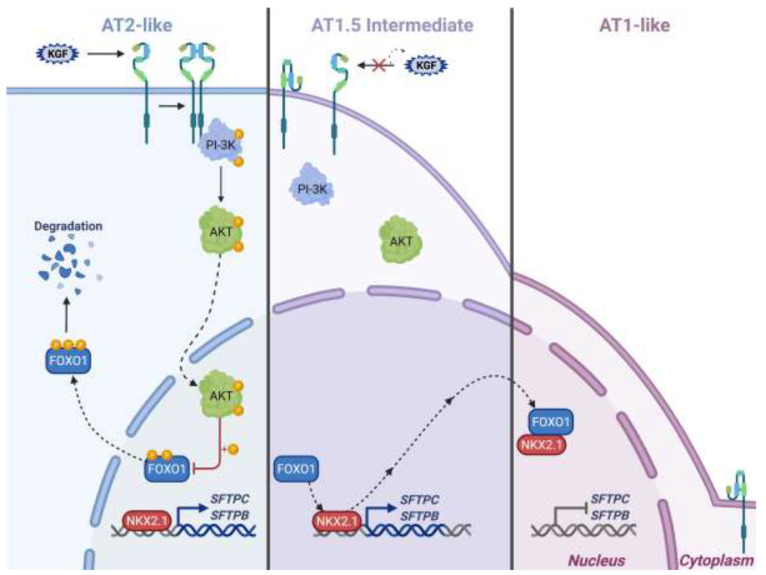
Model of mechanisms underlying FOXO1 regulation of lung-specific NKX2.1-activated genes via PI-3K/AKT-dependent phosphorylation. In AT2 and AT2-like cells in the presence of KGF, the PI-3K/AKT pathway is activated, leading to downstream phosphorylation of FOXO1 and subsequent cytoplasmic extrusion and degradation. The absence of nuclear FOXO1 allows active NKX2.1 to remain at the *SFTPC* and *SFTPB* promoters, increasing the expression of these AT2 cell-specific target genes. During the transition from AT2 to AT1-like cells (AT1.5 intermediate cells), or in the absence of KGF, the PI-3K/AKT pathway becomes inactivated. This in turn prevents the phosphorylation of FOXO1, allowing it to remain in the nucleus and interact with and pull NKX2.1 from its target promoters. This process is completed in differentiated AT1-like cells, where NKX2.1 removal from target genes by FOXO1 leads to the inhibition of gene expression. The figure was created with BioRender (BioRender Academic License https://biorender.com/ (accessed on 14 March 2022)).

## Data Availability

GSM4795154, GSM4795155), GSM47951551 and GSM7955150) from the Gene Expression Omnibus (GSE158196).

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
