# Peer review of "FOXO1 Couples KGF and PI-3K/AKT Signaling to NKX2.1-Regulated Differentiation of Alveolar Epithelial Cells"

_cells, 2022, doi:10.3390/cells11071122_

Round 1

Reviewer 1 Report

In this study by Zhong et al., the authors investigate and propose a coupled transcriptional/signaling model for differentiation of Alveolar Cell Culture. More specifically, they identify that master regulator NKX2.1, which promotes surfactant gene expression in AT2 cells, is inhibited by FOXO1 expression - leading to loss of expression, and promoting differentiation into the AT1 cell-type. Through experiments with different protein isoforms (+/- multiple domains), they potentially narrow down the basis of this inhibition to direct protein-protein interactions between FOXO1 and NKX2.1 that is independent of FOXO1's DNA-binding ability. Finally, they identify, through transfection experiments, that phosphorylation (in a PI-3K/AKT-signaling dependent manner) of FOXO1 prevents interaction with NKX2.1. Thus, suppression of AKT-mediated phosprylation promoted FOXO1-NKX2.1 association, promoting cell differentiation.

Overall, this is a well-written paper and the results are cleanly presented and I recommend publication. I suggest some minor concerns that would improve presentation/clarity of the paper.

  1. For the domain-mutant experiments to isolate mode of interaction, are concentrations of fusion proteins similar across conditions? If not, there is a concern you may be titrating against binding affinities and not reporting absolute presence/absence of interactions.
  2. The authors also notice an increase in FOXO3 expression (Fig 1.) - do you think these act as redundant paralogs for AT1 differentiation? Can the authors comment on this topic in their manuscript?
  3. There are previous reports of de-differentiation from AT1-like cells to AT2-cells. Can the authors comment on whether they believe their signaling network may be reversible i.e. whether upregulation of AKT signaling may promote AT2 formation? Including this topic in discussion may be of interest.

Author Response

  1. For the domain-mutant experiments to isolate mode of interaction, are concentrations of fusion proteins similar across conditions? If not, there is a concern you may be titrating against binding affinities and not reporting absolute presence/absence of interactions.

We appreciate the Reviewer’s comment. In our experiments, equal volumes of GST fusion protein-Sepharose beads as well as equal amounts of prey proteins from the TNT reaction were used (in the same reaction volume) across different conditions/reactions. To make sure similar amounts of different GST-fusion proteins were used, we first examined the expression levels of the different fusion proteins in bacteria using Western blotting analysis (Methods, section 2.7). Based on the results (Supplemental Fig. S3), we adjusted the amount of bacterial culture to be used for immobilizing similar amounts of GST-fusion proteins onto GST-Sepharose beads.

  1. The authors also notice an increase in FOXO3 expression (Fig 1.) - do you think these act as redundant paralogs for AT1 differentiation? Can the authors comment on this topic in their manuscript?

We replaced the Western blot image for FOXO3 with an image that is more representative of the quantitative analysis (Figure 1A), and now clarify by adding to the description line 333-334) “although FOXO3 shows a trend toward an increase on early days in culture (D1 and D3)”. We added in the Discussion (line 588-592) that “The trend of FOXO3 upregulation at the early stage of AT2-AT1 cell differentiation (Day 1 and 3) and stabilization at the later stages of differentiation (Day 5 and Day 8) suggest that FOXO3 may also be involved in the early AEC differentiation, although this did not reach statistical significance.” We also modified the sentence in discussion (line 592-594) to “It will be interesting to assess double and/or triple FOXO1/FOXO3/FOXO4 knockout mice to study the specific role of FOXO1 and the redundancy of FOXO family members (e.g., FOXO3) during AT2 to AT1 cell differentiation.”

3. There are previous reports of de-differentiation from AT1-like cells to AT2-cells. Can the authors comment on whether they believe their signaling network may be reversible i.e. whether upregulation of AKT signaling may promote AT2 formation? Including this topic in discussion may be of interest.

We added in the Discussion (line 654-662) that “We have previously reported that AEC treated with KGF from Day 4 to Day 8 reacquire lamellar bodies and demonstrate partial reversion to a cuboidal AT2 cell-like morphology suggesting that this signaling network may be reversible (6). In the current study, our results show that KGF and PI-3K/AKT regulate the maintenance of AT2 cell phenotype and AT2 to AT1 cell differentiation through FOXO1, which can be regulated by Ly294002, a PI-3K/AKT pathway inhibitor. Together, these studies suggest complex networks regulating AT2 to AT1 cell differentiation (69) and a role for PI-3K/AKT/FOXO1 pathway in the reprogramming of AT1 cell into AT2 cells following injury.”

Reviewer 2 Report

The manuscript describes FOXO1, other than FOXO3 or FOXO4, directly interacting with NKX2.1 to mediate ACE differentiation by interacting with the FK domain to interrupt NKX2.1 bind to SFTPC promoter through PI-3K/AKT pathway. Overall, the manuscript is well-written and well-prepared in most parts. I listed the following marks that the authors may want to consider to make them clearer before the further processing for publication.

Major Concerns and Comments:

  1. Line 3: I’d highly recommend the authors use the full name of “alveolar epithelial cell” in the title instead of “AEC”.
  2. Line 123: Please provide some more information about the rat age, I know you mentioned adult. Do you mean any age of adult rat can be used in this study?
  3. Line 333 and Figure 1: In Figure 1A the FOXO3 seems no different, while in Figure 1C, the similar increased expression trend with FOXO1 just without statistics difference as the provided result. Would you please provide some more information about this? Or do you think the lowest points in Figure 1C at D5 and D8 were due to the sample quality or some other reason?
  4. Lines 349-350: How can you conclude “FOXO1 inhibits NKX2.1-mediated SFTPC mRNA transcription/SFTPC transcriptional activity” from the results “knockdown of FOXO1 did not affect NKX2.1 expression”? I know Figures 2A and 2B are evidence of transcriptional activity. How did you exclude the translation activity inhibition?
  5. Line 354 and Figure S2: Nthy-ori 3-1 cell line, human thyroid epithelial cell line, has been used but no information about this cell line in the context and method part. I’d suggest the authors add some necessary information at least in the methods section.
  6. Lines 354-355: In line 111 the authors described the interaction of FOXO1 with NKX2.1 is cell-specific. There is only one more cell-type/tissue cell that had been tested, the human thyroid epithelial cell, instead of MLE-15. The authors concluded “FOXO1 repression of target genes is cell-type and tissue specific” is too wide. Have you ever tested other target genes other than SFTPC/SFTPB? Have you ever tested any other cell types/tissues instead of the two listed above?
  7. Line 430: Provide the reason why D3 was chosen instead of D5 and D8.
  8. Lines 646-647: The provided data only suggested: “KGF may plan an important role in AEC differentiation…”. Some further experiments need to be carried out if the authors would like to involve survival and proliferation.
  9. Please check throughout the whole manuscript to homogenous the format name for the genes and proteins.

Minor Concerns and Comments:

  1. Line 96: The “AEC” refers to “alveolar epithelial cell” has been defined two times, line 31 in the abstract and line 47. There is not necessary to define it again.
  2. Line 133: Please include antibody dilution ratio.
  3. Line 180 Western analysis part, line 191 co-IP part, and line 223 GST pull-down assay part: Please include antibody dilution ratio.
  4. Line 223 GST pull-down assay part: The font size in this part is quite different from other parts of the manuscript.
  5. Line 247: There is an extra underline “_” in the “4ºC”.
  6. Line 269: I’d suggest the authors define “RT” in line 187 from where the “room temperature” is first time recorded in the context.
  7. Line 309: I’d suggest the authors use italic p as it refers to a p-value. Please check throughout the whole manuscript.
  8. Line 309: I’d suggest the authors use “t-tests” or “t-test” instead of “t tests”.
  9. Line 314 Figure 1: The resolution of Figure 1E is too low. Please provide a higher resolution image.
  10. Line 320: Please check the format, “FoxO1”.
  11. Line 328: “Western analysis” should be “Western blotting analysis”, or “Western blot analysis”.
  12. Line 357 Figure 2: The “*” in Figure 2B and 2F are not in the right position.
  13. Line 675: Please cited the software or online tool resource for the supporting of Figure 6 image generation.
  14. Extra space check: line 52 between “as” and “primary”; line 62: before the last word of this line; line 299: before “MLE-15”; line 440: there is an extra space followed “protein contains”; line 479: there is an extra space before the word “together”; line 589: there seems an extra space followed the word “critical”; line 596: there is an extra space followed “stem cells”; line 701: there is an extra space followed the word “manuscript”; Please check throughout the whole manuscript.
  15. Please check the online instructions of [Cells] and make necessary changes for publication, especially the citations.

Author Response

Major Concerns and Comments:

1. Line 3: I’d highly recommend the authors use the full name of “alveolar epithelial cell” in the title instead of “AEC”.

“AEC” has been replaced with “alveolar epithelial cells”

2. Line 123: Please provide some more information about the rat age, I know you mentioned adult. Do you mean any age of adult rat can be used in this study?

We added “125- to 150 g” to clarify the adult rats used in the experiments.

3. Line 333 and Figure 1: In Figure 1A the FOXO3 seems no different, while in Figure 1C, the similar increased expression trend with FOXO1 just without statistics difference as the provided result. Would you please provide some more information about this? Or do you think the lowest points in Figure 1C at D5 and D8 were due to the sample quality or some other reason?

We replaced the Western blot image for FOXO3 with an image that is more representative of the quantitative analysis (Figure 1A), and now clarify by adding to the description (now line 333-334) “although FOXO3 shows a trend toward an increase on early days in culture (D1 and D3)”. We added in the Discussion (line 588-592) that “The trend of FOXO3 upregulation at the early stage of AT2-AT1 cell differentiation (Day 1 and 3) and stabilization at the later stages of differentiation (Day 5 and Day 8) suggest that FOXO3 may also be involved in the early AEC differentiation, although this did not reach statistical significance.” We also modified the sentence in discussion (line 592-594) to “It will be interesting to assess double and/or triple FOXO1/FOXO3/FOXO4 knockout mice to study the specific role of FOXO1 and the redundancy of FOXO family members (e.g., FOXO3) during AT2 to AT1 cell differentiation.”

4. Lines 349-350: How can you conclude “FOXO1 inhibits NKX2.1-mediated SFTPC mRNA transcription/SFTPC transcriptional activity” from the results “knockdown of FOXO1 did not affect NKX2.1 expression”? I know Figures 2A and 2B are evidence of transcriptional activity. How did you exclude the translation activity inhibition?

Thank you for raising this point. We examined NKX2.1 expression to exclude the possibility that FOXO1 was exerting its effects on SFTPC mRNA expression by decreasing NKX2.1 expression. Since NKX2.1 expression did not change, we concluded that FOXO1 inhibited NKX2.1-mediated SFTPC mRNA transcription. However, we cannot exclude translation activity inhibition and have modified the sentence (now line 351-352) to “FOXO1 inhibits SFTPC expression at least in part via reducing NKX2.1-mediated SFTPC transcriptional activity, although we cannot exclude inhibition of translational activity.”

5. Line 354 and Figure S2: Nthy-ori 3-1 cell line, human thyroid epithelial cell line, has been used but no information about this cell line in the context and method part. I’d suggest the authors add some necessary information at least in the methods section.

In Methods in the cell culture section 2.2 (line 141-142), we added: “Nthy-ori 3-1 cell line (#90011609, Millipore, Sigma) was cultured in RPMI 1640 medium supplemented with 2 mM glutamine and 10% FBS.”  

6. Lines 354-355: In line 111 the authors described the interaction of FOXO1 with NKX2.1 is cell-specific. There is only one more cell-type/tissue cell that had been tested, the human thyroid epithelial cell, instead of MLE-15. The authors concluded “FOXO1 repression of target genes is cell-type and tissue specific” is too wide. Have you ever tested other target genes other than SFTPC/SFTPB? Have you ever tested any other cell types/tissues instead of the two listed above?

We did not test any other cell types/tissues other than the two listed above and agree with the Reviewer’s comment of not being too broad in our conclusions. Accordingly, we have modified our conclusion (line 357) to “FOXO1 repression of target genes is AT2 cell specific.”

7. Line 430: Provide the reason why D3 was chosen instead of D5 and D8.

We clarified (line 433-434) that “on D3 from cultured primary rat AEC, a transitional time-point at which SFTPC expression was first absent, while FOXO1 expression is starting to increase (Figure 1A and 1B) and NKX2.1 expression was still present” (73).

8. Lines 646-647: The provided data only suggested: “KGF may plan an important role in AEC differentiation…”. Some further experiments need to be carried out if the authors would like to involve survival and proliferation.

We modified (now line 633-634) to “KGF may play an important role in maintaining AT2 cell homeostasis via PI-3K/AKT-mediated downregulation of FOXO1 activity during repair following lung injury.”

9. Please check throughout the whole manuscript to homogenous the format name for the genes and proteins.

We have checked throughout the manuscript for consistency of gene and protein name/formats.

Minor Concerns and Comments:

1. Line 96: The “AEC” refers to “alveolar epithelial cell” has been defined two times, line 31 in the abstract and line 47. There is not necessary to define it again.

We have corrected this and removed the definition from line 47.

2. Line 133: Please include antibody dilution ratio.

We have added “with a dilution ratio of 1:1000”

3. Line 180 Western analysis part, line 191 co-IP part, and line 223 GST pull-down assay part: Please include antibody dilution ratio.

Antibody dilution or amount were clarified in “Materials and Methods” sections 2.4, 2.6 and 2.8

4. Line 223 GST pull-down assay part: The font size in this part is quite different from other parts of the manuscript.

We have corrected the font throughout.

5. Line 247: There is an extra underline “_” in the “4ºC”.

We have deleted this extra underline.

6. Line 269: I’d suggest the authors define “RT” in line 187 from where the “room temperature” is first time recorded in the context.

We have defined RT as suggested.

7. Line 309: I’d suggest the authors use italic p as it refers to a p-value. Please check throughout the whole manuscript.

“P<0.05” was changed into “P<0.05” throughout the manuscript.

8. Line 309: I’d suggest the authors use “t-tests” or “t-test” instead of “t tests”.

We have changed “t tests” to “t-tests”.

9. Line 314 Figure 1: The resolution of Figure 1E is too low. Please provide a higher resolution image.

A higher resolution images is now provided in Figure 1E.

10. Line 320: Please check the format, “FoxO1”.

This is for rat shRNA, so “FoxO1” is correct.

11. Line 328: “Western analysis” should be “Western blotting analysis”, or “Western blot analysis”.

“Western analysis” has been changed to “Western blotting analysis”.

12. Line 357 Figure 2: The “*” in Figure 2B and 2F are not in the right position.

“*” in Figure 2B and 2F were adjusted to be in the right position.

13. Line 675: Please cited the software or online tool resource for the supporting of Figure 6 image generation.

We have added “Figure was created with BioRender.com” in the figure legend.

14. Extra space check: line 52 between “as” and “primary”; line 62: before the last word of this line; line 299: before “MLE-15”; line 440: there is an extra space followed “protein contains”; line 479: there is an extra space before the word “together”; line 589: there seems an extra space followed the word “critical”; line 596: there is an extra space followed “stem cells”; line 701: there is an extra space followed the word “manuscript”; Please check throughout the whole manuscript.

The manuscript has been checked and extra spaces have been deleted.

15. Please check the online instructions of [Cells] and make necessary changes for publication, especially the citations.

All reference brackets in the main text have been changed to square brackets in main text: (1) -> [1].

Round 2

Reviewer 2 Report

The authors took their efforts to fix the comments from the reviewers. I have no further concerns about this manuscript now.